# Nocardia Infections in the Immunocompromised Host: A Case Series and Literature Review

**DOI:** 10.3390/microorganisms10061120

**Published:** 2022-05-29

**Authors:** Emanuele Palomba, Arianna Liparoti, Anna Tonizzo, Valeria Castelli, Laura Alagna, Giorgio Bozzi, Riccardo Ungaro, Antonio Muscatello, Andrea Gori, Alessandra Bandera

**Affiliations:** 1Infectious Diseases Unit, Fondazione IRCCS Ca’ Granda Ospedale Maggiore Policlinico, 20122 Milano, Italy; arianna.liparoti@unimi.it (A.L.); anna.tonizzo@unimi.it (A.T.); valeria.castelli@unimi.it (V.C.); laura.alagna@policlinico.mi.it (L.A.); giorgio.bozzi@policlinico.mi.it (G.B.); riccardo.ungaro@policlinico.mi.it (R.U.); antonio.muscatello@policlinico.mi.it (A.M.); andrea.gori@unimi.it (A.G.); alessandra.bandera@unimi.it (A.B.); 2Department of Pathophysiology and Transplantation, University of Milano, 20133 Milano, Italy

**Keywords:** nocardia, nocardiosis, immunocompromised, transplant, haematologic malignancies, SOT, HSCT

## Abstract

Nocardia is primarily considered an opportunistic pathogen and affects patients with impaired immune systems, solid-organ transplant recipients (SOTRs), and patients with haematologic malignancies. We present the cases of six patients diagnosed with nocardiosis at our center in the last two years, describing the various predisposing conditions alongside the clinical manifestation, the diagnostic workup, and the treatment course. Moreover, we propose a brief literature review on Nocardia infections in the immunocompromised host, focusing on SOTRs and haematopoietic stem cell transplantation recipients and highlighting risk factors, clinical presentations, the diagnostic tools available, and current treatment and prophylaxis guidelines.

## 1. Introduction

*Nocardia* species are ubiquitous, filamentous, aerobic, Gram-positive, partially acid-fast bacilli that can cause localized or disseminated disease, most commonly affecting the lungs, skin and brain [1]. Over 50 species are known and fit into six major taxa (*N. abscessus*, *N. nova complex*, *N. farcinica*, *N. brevicatena/N.paucivorans*, *N. transvalensis complex*, and *N. cyriacigeorgica*), though only 12 to 16 are known to cause infections in humans [2]. Infection usually affects the lungs because the main portal of entry is by inhalation; however, cutaneous involvement (by inoculation) is not rare and it can disseminate to other organs, particularly to the central nervous system (CNS).

*Nocardia* can seldom cause infections in immunocompetent subjects, but it is primarily considered an opportunistic pathogen, more commonly affecting patients with impaired immune systems such as solid-organ transplant recipients (SOTRs) and patients with haematologic malignancies [3,4,5,6,7,8,9].

In our report, we present the cases of six immunocompromised patients diagnosed with nocardiosis, describing the different manifestations and analyzing the predisposing conditions and their clinical course.

## 2. Materials and Methods

We retrospectively analyzed the clinical records of adult patients diagnosed with infection due to *Nocardia* spp. (a culture of the causative organism from the infection site) during the period of 2020–2022 at the IRCCS Ca’ Granda Ospedale Maggiore Policlinico Foundation, a research and teaching hospital with 900 beds comprising a Bone Marrow Transplant unit and Solid-Organ Transplant units (liver, lung, kidney). All the cases with microbiologic confirmation occurring in the study period extracted from the infectious diseases consultation database were included

Microbiological diagnosis was obtained through standard culture of the selected samples.

The study was registered by the Milan Area 2 Ethical Committee (number 454_2022, approval Date: 10 May 2022).

## 3. Cases

### 3.1. Patient 1

A 60-year-old woman received a bilateral lung transplant (LT) in 2019, starting immunosuppressive therapy with tacrolimus (1.5 mg/day) and prednisone (17.5 mg/day). She was not receiving TMP/SMX primary prophylaxis. Five months after transplantation, the patient was admitted to our hospital for fever, nausea, dyspnoea and asthenia. Empirical therapy with levofloxacin and piperacillin/tazobactam was started. A chest CT scan showed pulmonary thromboembolism, nodular lesions in the upper left lobe and lower right lobe, and disseminated millimetric nodular lesions in the entire right hemithorax. A blood culture and bronchoalveolar lavage (BAL) were performed, the latter showing positivity for galactomannan antigen (index 1.00). Two days after hospital admission, the patient developed drowsiness, left-leg weakness, and left-facial-nerve deficit. A brain CT scan detected a subarachnoid haemorrhage in the frontal right lobe, suggestive of a mycotic aneurysm rupture. Due to these findings, endocarditis was suspected, and therapy was shifted to ceftriaxone (2 g iv every 12 h) plus vancomycin (subsequently shifted to linezolid, 600 mg iv every 12 h, due to acute kidney injury) plus amphotericin B (5 mg/kg iv every 24 h), added in relation to risk factors and preliminary microbiological results. Furthermore, the patient underwent vascular embolization, complicated by cerebral artery embolic occlusion. Seven days after specimen collection, the microbiology laboratory detected the growth of *Nocardia farcinica* in the blood samples and BAL. The antibiotic therapy was shifted to meropenem (2 g iv every 12 h) plus linezolid (600 mg iv every 12 h). The clinical conditions progressively worsened, and the patient died 8 days after hospital admission.

### 3.2. Patient 2

A 57-year-old man received orthotopic liver transplantation (OLT) for hepatitis B-virus-related cirrhosis, alcohol abuse, and multifocal hepatocellular carcinoma (HCC) in 2018. Immunosuppressive treatment with tacrolimus was started. One year later, the patient was diagnosed with granulomatosis with polyangiitis (formerly known as Wegener’s granulomatosis), and polymyalgia. Steroid therapy was administered (methylprednisolone, 1 g iv for 3 days followed by prednisone, 1 mg/kg per os). Prophylaxis with TMP/SMX (160/800 mg, 1 tab, 3 times a week) was started. Four years after OLT, the patient was admitted to our hospital for a painful swelling in the left thigh associated with fever. Computed tomography (CT) angiography of the left leg showed two hypodense masses compatible with organizing hematoma. The lesions were surgically drained, with discharge of purulent material, and empirical antibiotic treatment with piperacillin/tazobactam was started. Cultural analysis of the specimen resulted in *Nocardia nova* spp. *Africana* isolation. A brain CT scan showed an abscess in the right parietal lobe. Targeted antibiotic therapy with ceftriaxone (2 g iv every 12 h) plus TMP/SMX (15 mg/kg/day iv divided into three doses) was started and continued for 6 weeks, with a subsequent oral switch to TMP/SMX (800/160 mg, two tabs every 12 h) plus minocycline (200 mg every 12 h). Follow-up imaging (a musculoskeletal ultrasound and brain CT scan) showed progressive improvement in the lesions. The patient is still on treatment (12 months to-date) and is undergoing periodic clinical monitoring.

### 3.3. Patient 3

A 61-year-old woman with a history of primary biliary cirrhosis (PBC) and autoimmune hepatitis on a waiting list for OLT, who was being treated with corticosteroids in the last 5 months and was not in primary prophylaxis with TMP/SMX, was hospitalized due to altered mental status and asthenia. A brain CT scan was negative for active lesions and a chest CT scan showed several excavated bilateral pulmonary lesions, despite no clinical signs of pulmonary involvement. Cultures performed on blood and BAL specimens were negative. Due to a progressive improvement in their clinical conditions, the patient was discharged, and then readmitted two weeks later due to drowsiness. A new chest CT scan was performed, confirming the presence of excavated lesions; it also detected parenchymal consolidations with an aerial bronchogram and ground-glass opacities. A biopsy of the excavated lesions was deemed not feasible. Empirical antibiotic treatment with piperacillin/tazobactam was started. BAL was performed again with isolation of *Aspergillus fumigatus*, *Pneumocystis jirovecii* and *Nocardia nova complex*. Due to these findings, and according to AST, therapy with isavuconazole (200 mg iv every 8 h for 48 as a loading dose, then 200 mg iv every 24 h) and TMP/SMX (15 mg/kg iv divided into three doses) was started; TMP/SMX was later stopped due to myelosuppression. Given the lack of clinical improvement, therapy was then switched to imipenem/cilastatin (500 mg iv every 8 h) plus ceftobiprole (500 mg iv every 12 h) plus amphotericin B (150 mg iv every 24 h). A new chest CT scan was acquired, showing worsening of the parenchymal consolidations and of the ground-glass pattern, with development of a left pleural effusion. For suspected *Pneumocistys jirovecii* pneumonia, TMP/SMX was reintroduced along with prednisone (40 mg every 12 h). The clinical condition further deteriorated, and the patient died 30 days after hospital admission.

### 3.4. Patient 4

A 61-year-old man with a history of ischemic cardiomyopathy, hypertension, and bladder cancer, was diagnosed with chronic renal failure in IgA nephropathy (Berger’s disease). He started treatment with a steroid pulse regimen (also known as a “Pozzi scheme”, 1 g/day of methylprednisolone pulses for 3 consecutive days at the beginning of months 1, 3, and 5, followed by 0.5 mg/kg of oral prednisolone every other day) and, subsequently, mycophenolate as a chronic immunosuppressive therapy. The patient did not receive primary prophylaxis with trimethoprim-sulfamethoxazole (TMP/SMX). Five years after the diagnosis, he was admitted to our hospital for asthenia and decompensated diabetes. During hospitalization, a Positron Emission Tomography/Computed Tomography (PET/CT) scan was performed, showing increased fluorodeoxyglucose (FDG) uptake in a lung consolidation and in the right arm muscles (Figure 1B). A musculoskeletal ultrasound confirmed the presence of multiple hypoechogenic formations in the biceps brachii, compatible with intramuscular abscesses. Needle aspiration biopsy was performed, with isolation of Nocardia farcinica. Microbiological data became available after the patient’s discharge. The presence of cerebral abscesses was excluded by brain magnetic resonance imaging (MRI) and, because of the stable clinical conditions of the patient, antibiotic therapy with TMP/SMX (160/800 mg, two tabs every 12 h) was started with outpatient follow-up. The treatment was subsequently switched to linezolid (600 mg every 12 h) due to renal toxicity after one week. Association therapy with another drug (such as an aminoglycoside) was considered but not administered, since it was contraindicated by our consultant nephrologist. Antibiotic treatment was stopped after 5 months, given the resolution of the lesion at follow-up lung CT scan, the stable clinical conditions, and the discontinuation of immunosuppressive therapy.

### 3.5. Patient 5

An 80-year-old man with a history of Waldenstrom macroglobulinemia progressed to lymphoplasmacytic lymphoma and was placed on treatment with ibrutinib (420 mg/day) plus acyclovir primary prophylaxis. After eleven months, he was admitted to the emergency room due to progressive ideomotor impairment, disorientation, and memory deficiency. A brain CT scan and brain MRI showed necrotic lesions with associated oedema in the left frontal and parietal lobe (Figure 1C) and in the left cerebellum. A diagnostic brain biopsy was performed, with drainage of purulent material. Ibrutinib treatment was stopped, and empirical therapy was started with CEP (2 g iv every 8 h) plus vancomycin (1 g iv loading dose followed by 2 g every 24 h). Ten days later Bacillus cereus and *Nocardia wallacei* were isolated from a culture performed on the brain specimen. According to antibiotic susceptibility testing (AST), therapy was shifted to TMP/SMX (15 mg/kg iv divided into three doses) plus linezolid (600 mg iv every 12 h). A chest CT scan was performed, detecting multiple bilateral nodules and consolidations, along with bilateral pleural effusion. These findings were compatible with *Nocardia* infection. Subsequent follow-up imaging (chest CT scan and brain MRI) showed progressive improvement of the lesions. After six weeks of intravenous therapy, the antibiotic course was continued with a switch to minocycline (100 mg every 12 h) plus TMP/SMX (160/800 mg, two tabs every 12 h). The patient is still in treatment (12 months to-date) and in follow-up.

### 3.6. Patient 6

A 55-year-old man was diagnosed with Sezary syndrome (stage IV B) and started treatment with prednisone (prednisone higher than 20 mg for at least 6 months) and gemcitabine (four cycles), and then with mogamulizumab (two cycles, four and five months after the diagnosis). Prophylaxis with TMP-SMX (160/800 mg, one tab 3 times a week) was started. For disease progression, he was administered alemtuzumab (two cycles, seven and eight months after the diagnosis) and he developed cytomegalovirus (CMV) reactivation as a complication of alemtuzumab-induced lymphopenia. Nine months after diagnosis, the patient was admitted to our hospital for mild coronavirus disease 19 (COVID-19) pneumonia, treated with dexamethasone (6 mg iv for ten days) and low-flow oxygen, while he was on TMP/SMX primary prophylaxis. Ten days after discharge, he was readmitted to our hospital for fever, mild dyspnoea, and elevated inflammatory markers. A chest CT scan showed a consolidation (56 × 47 mm) in the left upper lobe. BAL was performed and empirical therapy with piperacillin/tazobactam plus voriconazole was started. Five days after collection, *Nocardia abscessus* was isolated from the BAL specimen. Antibiotic therapy with TMP-SMX (15 mg/kg/day iv, divided into three doses) plus ceftriaxone (2 g iv every 12 h) was started. A total-body CT scan was performed, excluding other disease localization. Prompt clinical and radiological response were achieved, with stable apyrexia and resolution of the lung lesion at a subsequent chest CT scan (Figure 1A). The patient is still in treatment (2 months to-date) and clinical follow-up.

## 4. Literature Review

*Nocardia* spp. infection mainly affects subjects with impaired immune systems [3]. More than 60% of the reported cases of nocardiosis are associated with at least one of the following conditions: solid-organ transplant (SOT), haematopoietic stem-cell transplantation (HSCT), human immunodeficiency virus (HIV) infection with CD4 counts of <100 cells/mm^3^, hematologic malignancy, and chronic corticosteroid therapy [10].

The frequency of *Nocardia* infections in solid-organ transplant recipients varies between <1% and 3.5%, according to different reports concerning distinct types of transplant [11]. A matched case–control study comprising 5126 SOTRs demonstrated a 3.5% rate of infection among lung transplant recipients, with rates in heart, bowel, kidney, or liver recipients of 2.5%, 1.3%, 0.2%, and 0.1%, respectively [12]. Among SOTRs, lung recipients are the ones at major risk [4]. This can be explained by the increase in these procedures over time, the high levels of immunosuppression needed, and the unique risk of direct exposure of the allograft to the outside environment with an organism mostly acquired via inhalation [5].

Nocardia infection rarely occurs within the first month after SOT, developing mostly within the first 1–2 years post-transplant [9,12]. Later onset is not uncommon, with a median time to infection from SOT of 34–38 months [6]; moreover, it is often temporally associated with intensified immunosuppression, such as anti-lymphocyte globulin administration or the use of a higher dosage of calcineurin inhibitors and corticosteroids [9,12].

Similarly to SOTRs, the incidence of nocardial infection in allogeneic HSCT recipients ranges from 1.7 to 2.6% [7,13,14]. The risk factors in this population are the administration of non-myeloablative regimen pre-HSCT (such as alemtuzumab, melphalan, and fludarabine) and prior graft-versus-host disease (GvHD) needing higher levels of immunosuppression.

The prevalence of the *Nocardia* species appears to vary by geography [10]. Depending on the setting, the most commonly identified species of *Nocardia* are *N. nova complex*, *N. farcinica*, *N. cyriacigeorgica*, *N. abscessus complex*, and *N. brasiliensis* [7,12,15].

*Nocardia* spp. can infect virtually all body sites. The lung is the most common site of infection, with CT scans showing lung nodules in around 75% of the cases [8]. More than 40% of nocardiosis presents as clinically disseminated disease, with CNS, skin, and soft-tissue involvement [8]. Cerebral abscesses are the most-reported presentation of CNS localization. Bacteraemia is rarely seen [2,16]. A recent systematic literature review [16] showed that isolation of *Nocardia* spp. on blood culture was strongly associated with the administration of immunosuppressive drugs (90% of the cases, mostly corticosteroids), the use of endovascular devices, haematological malignancy, SOT, and HSCT. The median incubation time to detection of *Nocardia* bacteraemia was 4 days.

According to the latest guidelines from the *American Society of Transplantation* [10], treatment options for empiric therapy in SOTRs include TMP/SMX, associated with a second agent (imipenem or amikacin) in the case of cerebral involvement or disseminated disease. Imipenem, ceftriaxone, or linezolid are alternative options for the first-line treatment of *Nocardia* infections in those with an allergy to TMP/SMX. However, it should be emphasized that there is an absence of consensus regarding the optimal empirical treatment of nocardiosis, with initial therapy usually selected based on the severity of clinical presentation, on the organs affected, and on local epidemiology. Definite treatment should be based on AST, with TMP/SMX as the first choice, used alone or as the backbone of a multi-drug regimen; in addition, linezolid was recently proposed as alternative for induction therapy [17]. The duration of therapy is still debated, and it should be tailored to the patient’s response to treatment and their clinical history (particularly on immune status and the concomitant use of immunosuppressants). Pulmonary and soft-tissue infections should be treated for at least 6 months, and disseminated disease for 6 to 12 months; parenteral therapy should be continued for 3 to 6 weeks; and clinical improvement should be seen before switching to an oral regimen. Notably, there are data suggesting that shorter courses of therapy (<120 days) may be successfully applied in isolated skin and soft-tissue infections, regardless of the status of the immune system [11]. Mortality rate is heterogeneous and dependent on different factors, such as a patient’s underlying condition and follow-up duration. Given the paucity of data and lack of studies, it is difficult to ascertain the attributable mortality rate; however, in a case–control study among SOT patients, those with nocardiosis had a significantly higher 6-month mortality rates than the controls patients (14% vs. 4%) [12]. Among HSCT patients, studies reported a mortality of 66%, with an attributable mortality ranging from 16 to 33% [11,14].

TMP-SMX prophylaxis (one double-strength tablet daily or three times weekly) may be helpful in preventing primary *Nocardia* infection or relapse after treatment, although infections can occur despite prophylaxis. Over the last decade, studies focusing on SOTRs already on TMP/SMX prophylaxis for a different reason (such as the prevention of *Pneumocystis jirovecii* pneumonia) showed that, in this population, 18% to 21% of nocardiosis cases were “breakthrough infections” [1,9]. A recent study on heart transplant recipients reported that 40% of cases occurred in patients on primary prophylaxis with TMP/SMX [18]. The lack of efficacy in preventing *Nocardia* infection can be attributed to pharmacodynamic phenomena (suboptimal dosing, impaired absorption), but also to a rise in antimicrobial resistance, post-transplant immunosuppression, and other comorbidities that affect the patient’s immune response.

## 5. Discussion

*Nocardia* infections have increased in the last two decades, likely due to improved detection and identification methods and the expanding immunocompromised population.

In this work, we described a case series of six patients diagnosed with nocardiosis during a two-year period (2020–2022) at our tertiary center (Table 1). All patients were immunocompromised: two out of six (33%) were SOTRs (liver, lung), two (33%) were patients with autoimmune diseases needing corticosteroids and disease-modifying antirheumatic drugs (PBC, IgA nephropathy), and two (33%) were subjects with haematologic malignancies (Sezary syndrome, Waldenstrom macroglobulinemia). In SOTRs, nocardiosis developed 5 months (lung) and 45 months (liver) after the transplant.

Importantly, the observed lethality rate was 33%, and particularly associated with late diagnosis and treatment start.

Although the lungs are often the primary site of infection, disseminated infection, defined as involvement of at two least non-contiguous organs, is not uncommon in transplant recipients.

Along with typical respiratory symptoms such as shortness of breath, a dry or productive cough, and haemoptysis, it is important to look for other manifestations such as CNS symptoms (headache, seizure, meningitis, focal neurologic deficits) and skin lesions (ulcers, cellulitis, nodules, abscess) in patients with a history of SOT, HSCT, and/or use of immunosuppressive agents (such as corticosteroids and calcineurin inhibitors).

Among our patients, the most common manifestation was pulmonary nocardiosis (66%), even though dyspnoea was present at hospital admission only in two patients (33%). Half of the subjects suffered CNS involvement, one third showed soft-tissue abscesses, and only in one case (16%) was *Nocardia* spp. isolated from the blood specimen.

Upon clinical suspicion of nocardiosis, the diagnostic workup should comprise appropriate radiological examination. A chest CT scan is the gold standard for radiological diagnosis of pulmonary nocardiosis, but CT findings are diverse and non-specific, usually showing irregular nodular lesions that may cavitate. Diffuse consolidative or interstitial pneumonic infiltrates and pleural effusion can also be found. Brain MRI/CT should be performed in all cases of Nocardia infection, to exclude CNS involvement. Localized lesions (such as skin and soft tissues) should be analyzed according to the best diagnostic method available (ultrasound, CT scan). Interestingly, FDG-PET has recently been considered as a useful tool for the diagnosis and staging of nocardiosis by revealing unknown foci of dissemination and for guiding biopsy; furthermore, it has been deemed useful for monitoring treatment response, shortening treatment duration and ruling out relapses [19]. In one of our patients, FDG-PET was useful in guiding a diagnostic biopsy, resulting in microbiological diagnosis and in staging disease dissemination.

Microbiological detection in specimens obtained from the infected lesion is paramount to confirm the diagnosis and to guide the treatment, since speciation and AST can help choose the most effective antimicrobial therapy. Prolonged incubation of diagnostic specimens should be performed in suspected cases; therefore, communication with clinical microbiology laboratory personnel is recommended.

Treatment selection should initially be made on the speciation, and then tailored on the AST. A backbone therapy with TMP/SMX, associated with carbapenems or aminoglycosides in the case of disseminated disease, has been proposed in the absence of microbiological input.

“Breakthrough infections” in patients undergoing TMP-SMX prophylaxis for other reasons are described in the literature, with rates ranging from 18 to 40%. Notably, in our series, nocardiosis occurred in two patients (33%) who were on TMP/SMX prophylaxis at the time of infection.

## 6. Conclusions

Nocardiosis in the immunocompromised host is a challenging medical entity that has gained ground in the last two decades, given the empowered diagnostic tools available and the growing number of subjects exposed to immunosuppressing agents for various reasons. Prompt recognition is crucial to achieve the correct diagnosis and treat these patients adequately.

Our case series describes how *Nocardia* infection can manifest in different clinical scenarios and in subjects with various predisposing factors, highlighting the need for careful evaluation in patients with an impaired immune system who show respiratory and systemic symptoms, looking for radiological and microbiological signs of nocardiosis.

Further studies are needed to better characterize the subjects at major risk and to develop new evidence-based guidelines on treatment and prophylaxis (both primary and secondary).

## Figures and Tables

**Figure 1 microorganisms-10-01120-f001:**
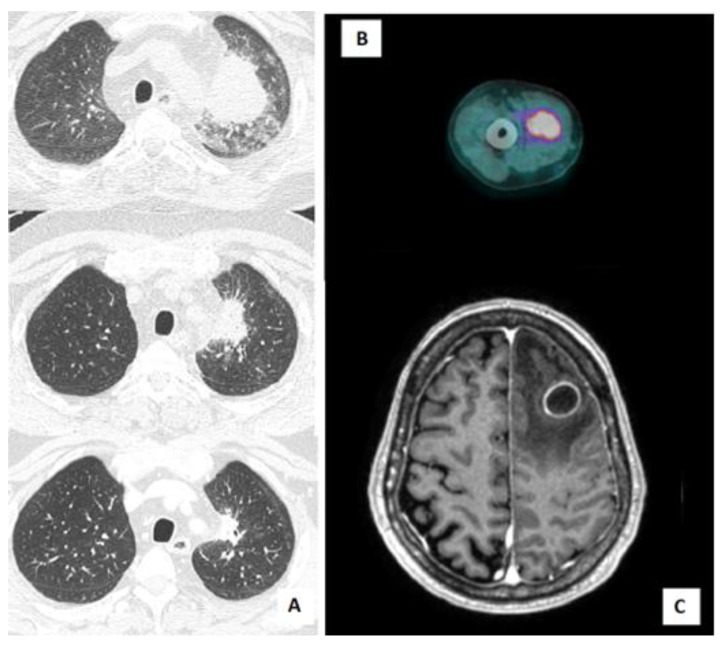
**Different radiological manifestations of Nocardia infection in immunocompromised hosts:** (**A**) Chest CT scan showing the evolution of lung lesions caused by *Nocardia abscessus* in patient 6 during antibiotic treatment (at diagnosis, after 2 weeks, and after 4 weeks); (**B**) PET/CT scan showing increased FDG uptake in the right arm muscles (*Nocardia farcinica* intramuscular abscess diagnosed in patient 4); (**C**) brain MRI showing left fronto-parietal abscess by *Nocardia wallacei* in patient 5.

**Table 1 microorganisms-10-01120-t001:** **Patients’ characteristics.** Pt: patient; CLSI: Clinical and Laboratory Standards Institute; TMP-SMX: trimethoprim-sulfamethoxazole; AMI: amikacin; LIN: linezolid; AMC: amoxicillin-clavulanate; CIP: ciprofloxacin; MIN: minocycline; CEP: cefepime; CTR: ceftriaxone; CLA: clarithromycin; IMI: imipenem; TOB: tobramycin; F/U: follow-up; MRI: magnetic resonance imaging; CT: computed tomography; HCC: hepatocellular carcinoma; COPD: chronic obstructive pulmonary disease; HBV: hepatitis B virus; iv: intravenous; po: per os; CMV: cytomegalovirus; BAL: bronchoalveolar lavage; AST: antibiotic sensibility testing; N/A: not available; PBC: primary biliary cirrhosis; OLT: orthotropic liver transplantation.

Pt	Gender, Age	Relevant Clinical History	Corticosteroids and Immunosuppressive Drugs	TMP-SMX Prophylaxis	Site(s) of Infection	Symptoms and Radiological Findings	Species of Nocardia	Antibiotic Susceptibility According to CLSI Breakpoints	Targeted Therapy	Duration of Treatment	Outcome and Follow Up
1	F, 60	Bilateral lung transplant recipient; CMV reactivation; steroid-induced diabetes	Chronic therapy with prednisone Tacrolimus	NO	Lung, brain Colture from BAL and blood	Fever, asthenia, dyspnoea, drowsiness, right-leg weakness, left-facial-nerve deficit Chest CT scan: upper left lobe and lower right lobe nodular lesions Brain CT scan: brain lesions; frontal right lobe subarachnoid haemorrhage; mycotic aneurysms of cerebral artery	*Nocardia farcinica*	S: AMI, AMC, CIP, IMI, LIN, MIN, TMP/SMX R: CEP, CTR, CLA, TOB	Not started due to patient’s death before AST result	N/A	Death
2	M, 57	Liver transplantation for alcohol and HBV-related cirrhosis, multifocal HCC; granulomatosis with polyangiitis, polymyalgia, steroid-induced diabetes, COPD	Three bolus methylprednisolone (1 g ev for 3 days) then chronic therapy with prednisone Tacrolimus	YES	Soft tissues, brain Colture from needle biopsy	Painful swelling in the left thigh, fever Lower limbs CT scan: left leg hematomas Brain CT scan: right parietal abscess	*Nocardia nova* spp. *africana*	S: AMI; CEP; CTR, CLA; IMI; LIN; MIN; TMP/SMX I: TOB R: AMC; CIP	CTR + TMP/SMX → TMP/SMX + MIN	6 weeks iv + 12 months po (ongoing)	Progressive improvement of the lesions F/U ongoing
3	F, 61	PBC and autoimmune hepatitis on waiting list for OLT; steroid induced diabetes	Chronic therapy with prednisone	NO	Lung Colture from BAL	Back pain, drowsiness, Chest CT scan: several excavated bilateral lesions; parenchymal consolidations with aerial bronchogram and ground-glass opacities	*Nocardia nova complex*	S: AMI, CEP, CTR, CLA, LIN, TMP/SMX I: IMI, MIN R: AMC, CIP, TOB	Ceftobiprole + IMI/cilastatin + TMP/SMX	N/A	Worsening of lung lesions with pleural effusion Death
4	M, 61	Chronic renal failure in IgA nephropathy, diabetes, ischemic cardiomyopathy, previous bladder cancer (F/U negative)	Three bolus methylprednisolone (1 g iv for 3 days) then chronic therapy with prednisone Previous treatment with mycophenolate (suspended 7 months before infection)	NO	Soft tissues Colture from needle biopsy	Asthenia PET/CT total body: enhanced captation of right arm muscles Musculoskeletal ultrasound: hypoechogenic formations in the biceps brachii Brain MRI: negative	*Nocardia farcinica*	S: AMI, LIN; TMP/SMX I: AMC; CIP; MIN R: CEP; CTR; CLA; IMI; TOB	TMP/SMX → LIN	5 months	Resolution of the lesion at F/U lung CT scan F/U ongoing
5	M, 80	Waldenstrom macroglobulinemia progressed to lymphoplasmacytic lymphoma; prostatic cancer in F/U	No corticosteroid therapy Ibrutinib	NO	Brain, lung Colture from brain biopsy	Confusion, drowsiness, right leg weakness, left facial nerve deficit Brain CT scan and MRI: left frontal and parietal lobe and left cerebellum necrotic/cystic lesions with associated edema Chest CT scan: several bilateral nodular infiltrates and parenchymal consolidations, bilateral pleural effusion	*Nocardia wallacei*	S: AMC, CTR, CIP, LIN, MIN, TMP/SMX I: CEP, CLA R: AMI, IMI, TOB	TMP/SMX → MIN	12 months (ongoing)	Progressive improvement of the brain and pulmonary lesions F/U ongoing
6	M, 55	Sezary syndrome, recent CMV reactivation	Chronic therapy with prednisone Gemcitabine (4 cycles), Mogamolizumab (two months), Alemtuzumab (two months)	YES	Lung Colture from BAL	Fever, cough Chest CT scan: consolidation in upper left lobe	*Nocardia abscessus*	S: AMI, AMC, CTR, LIN, TOB, TMP/SMX I: CIP, MIN R: CEP, CLA, IMI	TMP/SMX + CTR	4 months (ongoing)	Reduction of the lung lesion at subsequent chest CT scan F/U ongoing

## Data Availability

The data presented in this study are available within the article.

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
