# Peer review of "Nocardia Infections in the Immunocompromised Host: A Case Series and Literature Review"

_microorganisms, 2022, doi:10.3390/microorganisms10061120_

Round 1

Reviewer 1 Report

Figure 1B has an unexplained image which should be removed. It might be a PET-CT of a brain with ependymitis

The discussion should not  say that halo signs may be seen or that Thayer-Martin medium with antibiotics should be used.

The table is unnecessary.

Reviewer 2 Report

The work is devoted to the study of infections caused by the opportunistic pathogen Nocardia, in the immunocompromised host. 6 clinical cases are described and a brief but very clear and specific literature review on Nocardia infections in the immunocompromised host is made (focusing on solid organ transplant recipients and haematopoietic stem cell transplantation recipients). The literature review clearly and consistently analyzes the literature on Nocardia infections in the immunocompromised host. The experimental section clearly and consistently formulated the clinical picture of 6 patients. In addition, the key data are placed in Table 1, which allows the reader of the article to quickly and clearly operate on the presented results.

However, it is not clear on what basis patients 1-6 were sequentially placed. In the Discussion section, the authors make some classification of cases, indicating that all patients were immunocompromised: 2 out of 6 (33%) were SOTR (liver, lung), 2 (33%) were subjects with haematologic malignancies (Sezary syndrome, Waldenstrom macroglobulinemia), and 2 (33%) were patients with autoimmune diseases needing corticosteroids and disease modifying antirheumatic drugs (PBC, IgA nephropathy). But in Cases part of the article, patients are given separately: SOTR - 2 and 3 patients, haematologic malignancies - 4 and 6 patients, patients with autoimmune diseases needing corticosteroids and disease modifying antirheumatic drugs - 1 and 5. Order the observed subcases to make the material easier to comprehend.

Figure 1 does not have a common title, such as: "Diagnostic pattern of different cases of Nocardia infections in the immunocompromised hosts". It is impossible to understand from the caption to the figure whether these three illustrations refer to one patient, or to 2-3 independent cases. There is also no information whether one (or more) Nocardia species led to such manifestations diagnosed on CT, PET/CT and MRI. To understand what the A-C of the figure refers to, you need to read the entire Case part of the article. It turns out that 1A corresponds to the study for patient 6 (Nocardia abscessus), 1B corresponds to the study for patient 1 (Nocardia farcinica), 1C corresponds to the study for patient (Nocardia wallacei). Enter this information in the figure caption.

The authors indicate methods for diagnosing patients (for example, PET, CT). Specify the method(s) used to identify the species of microorganisms isolated from patient tissues, for example, adding it to the Materials and Methods section.

Comments:

Lines 265-266

As far as is clear from the description of the 5th patient, he was infected with a mixed infection and died of mycosis (suspected Pneumocistys jirovecii pneumonia). Is it correct to include this case in the mortality statistics (33% mortality, 2 cases out of 6) from Nocardia infections?

Lines 286-291

You indicate that FDG-PET has been recently considered as useful tool for diagnosis and staging of nocardiosis by revealing unknown foci of dissemination and for guiding biopsy, furthermore for monitoring treatment response, shortening treatment duration and rule out relapses. For what reason did FDG-PET not be done when diagnosing patients 2-6, but was done only for patient 1?

Lines 294-297

You indicate that the preliminary incubation of diagnostic specimens should be performed in suspected cases, along with use of selective media such as Thayer-Martin agar with antibiotics.

In connection with the fact, note in the Discussion section whether in your work for any Nocardia isolated from patients, microbiological analysis was carried out on agar media with antibiotics in order to select the most effective drug for treatment?   
